# Analysis of HIV/AIDS Epidemic and Socioeconomic Factors in Sub-Saharan Africa

**DOI:** 10.3390/e22111230

**Published:** 2020-10-29

**Authors:** Shuman Sun, Zhiming Li, Huiguo Zhang, Haijun Jiang, Xijian Hu

**Affiliations:** College of Mathematics and System Science, Xinjiang University, Urumqi 830046, China; Sunshuman2222@163.com (S.S.); zmli@xju.edu.cn (Z.L.); zhang_huiguo@163.com (H.Z.); jianghaixju@163.com (H.J.)

**Keywords:** HIV/AIDS epidemic, regression model, Newton–Raphson procedure, Fisher scoring algorithm, time series

## Abstract

Sub-Saharan Africa has been the epicenter of the outbreak since the spread of acquired immunodeficiency syndrome (AIDS) began to be prevalent. This article proposes several regression models to investigate the relationships between the HIV/AIDS epidemic and socioeconomic factors (the gross domestic product per capita, and population density) in ten countries of Sub-Saharan Africa, for 2011–2016. The maximum likelihood method was used to estimate the unknown parameters of these models along with the Newton–Raphson procedure and Fisher scoring algorithm. Comparing these regression models, there exist significant spatiotemporal non-stationarity and auto-correlations between the HIV/AIDS epidemic and two socioeconomic factors. Based on the empirical results, we suggest that the geographically and temporally weighted Poisson autoregressive (GTWPAR) model is more suitable than other models, and has the better fitting results.

## 1. Introduction

Acquired immunodeficiency syndrome (AIDS) is a malignant infectious disease with a high fatality rate caused by human immunodeficiency virus (HIV). The HIV/AIDS epidemic has been one of the greatest global public health and social development problems since 1981, particularly in Sub-Saharan Africa. As of 31 December 2016, over 30 million people had died from the disease [1]. More than 70% of the 35 million people are infected with the HIV/AIDS disease in Sub-Saharan Africa. Thus, the HIV/AIDS epidemic of Sub-Saharan Africa has attracted extensive attention from researchers around the world [2,3,4].

In earlier studies, Janet et al. [5] and Hallman et al. [6] demonstrated the relationship between the disease and socioeconomic status. Chris et al. [7] indicated socioeconomic factors to explain this disease outperformed cultural ones in South Africa. Mathematical models always play an important role in evaluating the trends of the HIV/AIDS epidemic [8]. For example, regression models have been widely used in the study of the relationship between this disease and influencing factors. Shiboski et al. [9] considered a generalized linear model to obtain the statistical analysis of the HIV/AIDS disease. A mixed-effects linear regression model was used to analyze the correlation between national population and antenatal care [10]. Laurence et al. [11] applied a spatial regression model to show that the epidemic had substantial geographic variance across Sub-Saharan Africa.

This paper proposes several regressive models to investigate the relationships between the HIV/AIDS epidemic, the gross domestic product (GDP) per capita and the population density in ten countries of Sub-Saharan Africa. The Poisson regression model is introduced in Section 2.1. Section 2.2 and Section 2.3 describe two spatial models, respectively. A spatiotemporal autoregressive model is proposed in Section 2.4. The maximum likelihood method is used to obtain the iterative formulas of coefficient estimations in Section 3. The main results are shown in Section 4, followed by discussion in Section 5.

## 2. Methodologies

### 2.1. Poisson Regression Model

Regression models are a set of statistical processes for estimating the relationships between response and explanatory variables. The classical model is a linear regression. Nelder and Wedderburn [12] extended the linear model to a generalized linear regression for solving the discrete data problem. This kind of models are very important in ecology, medicine and economics [13,14,15]. Suppose that $Y=(Y_{1},Y_{2},\ldots ,Y_{n})$ is the response variable, where $Y_{i}\left(i=1,\ldots ,n\right)$ are independent. The density function is
$$\begin{array}{c} f\left(y_{i};\theta_{i},\phi_{i}\right)=\exp\left(\frac{y_{i}\theta_{i}-b\left(\theta_{i}\right)}{a(\phi_{i})}+c\left(y_{i},\phi_{i}\right)\right), \end{array}$$
where a(·),b(·),c(·,·) are known functions, and $\theta_{i},\phi_{i}$ are unknown parameters for i=1,2,…,n. Denote $\mu_{i}=E\left(Y_{i}\right)$, and $g\left(\mu_{i}\right)=\ln\left(\mu_{i}\right)$ is a link function. Let $X_{ij}$ be explanatory variables for the *i*th observation in the *j*th variable. Then, the Poisson regression (PR) model is given by
(1)$$\begin{array}{c} g\left(\mu_{i}\right)≜\eta_{i}=\sum_{j=1}^{p}\beta_{j}X_{ij}, \end{array}$$
where i=1,2,…,n, and $\beta_{j}\left(j=1,2,\ldots ,p\right)$ are unknown parameters.

### 2.2. Geographically Weighted Poisson Regression Model

With in-depth study, regression models have been frequently applied in epidemiology and health geography for trying to investigate the persistent geographical variations in disease [16]. Based on the generalized linear regression, Brunsdon et al. [17] proposed the geographically weighted regression model to analyze the spatial non-stationary processes of discrete data. The disease maps arising from this process are considered through the establishment of the geographically weighted Poisson regression (GWPR) model [18,19,20] below
(2)$$\begin{array}{c} g\left(\mu_{i}\right)≜\eta_{i}=\sum_{j=1}^{p}\beta_{j}\left(u_{i},v_{i}\right)X_{ij}, \end{array}$$
where $\left(u_{i},v_{i}\right)\left(i=1,2,\ldots ,n\right)$ are the geographical locations, and $\beta_{j}\left(u_{i},v_{i}\right)\left(j=1,2,\ldots ,p\right)$ are unknown parameters at the position $\left(u_{i},v_{i}\right)$.

### 2.3. Geographically Weighted Poisson Autoregressive Model

Another issue deserving of special attention is whether there exists an interaction between different regions in terms of spatial data. Previous studies [21,22,23,24] showed that spatial data has not only spatial non-stationarity but also correlation. Zhang [25] proposed the geographically weighted Poisson autoregressive (GWPAR) model as follows:(3)$$\begin{array}{c} g\left(\mu_{i}\right)≜\eta_{i}=\rho \sum_{k=1}^{n}c_{ik}\eta_{k}+\sum_{j=1}^{p}\beta_{j}\left(u_{i},v_{i}\right)X_{ij}, \end{array}$$
where ρ is a scalar autoregressive parameter, and $c_{ik}\left(i,k=1,2,\ldots ,n\right)$ is the adjacency relation between the *i*th and *k*th locations. Let $c^{i}$ be the number of regions adjacent to the *i*th position. If the *k*th position is next to the *i*th’s, then $c_{ik}=1/c^{i}$. Otherwise, $c_{ik}=0$.

### 2.4. Geographically and Temporally Weighted Poisson Autoregressive Model

Recently, many spatiotemporal models have been proposed to describe the spatiotemporal variations in the relationships of response and explanatory variables [26,27]. Concerning the modeling of spatiotemporal data, there are two important properties: non-stationarity and auto-correlation. The non-stationarity indicates that there exists more than one linear relation between response and explanatory variables. It can be used to identify where interesting relationships are likely to occur or where detailed investigation is necessary in the study areas [28]. Spatiotemporal auto-correlation is an important factor to determine the temporal correlations of observations [29]. These two problems always appeared together [30]. A geographically and temporally weighted autoregressive (GTWPAR) model can be applied to account for non-stationary and auto-correlated effects simultaneously.

Let *Y* be the response variable, and $Y_{ik}\left(i=1,2,\ldots ,n_{k},k=1,2,\ldots ,T\right)$ be the independent variables of *Y* in the *i*th position and the *k*th time. The density function can be defined as follows:$$\begin{array}{c} f\left(y_{ik};\theta_{ik},\phi_{ik}\right)=\exp\left(\frac{y_{ik}\theta_{ik}-b\left(\theta_{ik}\right)}{a(\phi_{ik})}+c\left(y_{ik},\phi_{ik}\right)\right), \end{array}$$
where the parameters are similar to Section 2.1. Denote $\mu_{ik}=E\left(Y_{ik}\right)$, and $g\left(\mu_{ik}\right)=\ln\left(\mu_{ik}\right)$. Let $X_{ijk}\left(j=1,2,\ldots ,p\right)$ be the *j*th explanatory variable. The GTWPAR model is expressed by
(4)$$\begin{array}{c} g\left(\mu_{ik}\right)≜\eta_{ik}=\rho \sum_{m=1}^{T}\sum_{l=1}^{n_{k}}c_{lm}^{\left(ik\right)}\eta_{lm}+\sum_{j=1}^{p}\sum_{k=1}^{T}\beta_{jk}\left(u_{ik},v_{ik},t_{k}\right)X_{ijk}, \end{array}$$
where $\{\beta_{jk}\left(u_{ik},v_{ik},t_{k}\right)$} is a set of unknown parameters at the *i*th position in the *k*th time, and $c_{lm}^{\left(ik\right)}$ is the adjacent relation between the location $\left(u_{ik},v_{ik},t_{k}\right)$ and $\left(u_{lm},v_{lm},t_{m}\right)$. Following the work of [31], the spatiotemporal distance between the locations $\left(u_{ik},v_{ik},t_{k}\right)$ and $\left(u_{lm},v_{lm},t_{m}\right)$ can be defined as
$$d_{lm}^{\left(ik\right)}=\sqrt{\lambda \left[{\left(u_{ik}-u_{lm}\right)}^{2}+{\left(v_{ik}-v_{lm}\right)}^{2}\right]+\mu {\left(t_{k}-t_{m}\right)}^{2},}$$
where μ and λ are used to balance spatiotemporal distances. Suppose that
$$c_{lm}^{\left(ik\right)}=\left\{\begin{array}{cc} 1/c^{ik},\;\;\; & 0<d_{lm}^{\left(ik\right)}<d,\\ 0,\;\;\; & \mathrm{otherwise}, \end{array}\right.$$
where *d* is a constant and satisfies $\min\left\{d_{lm}^{\left(ik\right)}\right\}<d<\max\left\{d_{lm}^{\left(ik\right)}\right\}$.

Next, we rewrite the model (Equation 4) in a matrix form
$$\begin{array}{c} \eta =\rho C\eta +B^{\prime }X^{\prime }, \end{array}$$
where $\eta ={\left(\eta_{11},\cdots ,\eta_{n_{1}1},\eta_{12},\cdots ,\eta_{n_{2}2},\cdots ,\eta_{1T},\cdots ,\eta_{n_{T}T}\right)}^{{}^{\prime }}$, $C=(c_{lm}^{\left(ik\right)})$, $X=(X_{ijk})$ and $B=(\beta_{jk}\left(u_{ik},v_{ik},t_{k}\right))$. For convenience, define $\eta_{K}$ as the *K*th element of η; $C_{IK}$ and $X_{IK}$ are the *I*th row and the *K*th column of the matrices *C* and *X*, respectively. The detailed expressions of *C*, *X* and B are given in Section A.1.

**Remark** **1.**
*For the GTWPAR model (Equation 4), if ρ=0 and $\beta_{jk}\left(u_{ik},v_{ik},t_{k}\right)$ is independent of the spatiotemporal effect, the model is a PR model. If ρ=0 and $\beta_{jk}\left(u_{ik},v_{ik},t_{k}\right)$ is dependent on spatial effect but independent of temporal effect, the model becomes GWPR model. If ρ≠0 and $\beta_{jk}\left(u_{ik},v_{ik},t_{k}\right)$ is independent of temporal effect, it is the GWPAR model. Thus, PR, GWPR and GWPAR models are the special cases of the GTWPAR model.*


## 3. Coefficient Estimation

In this section, we only provide the estimation method of the GTWPAR model since the PR, GWPR and GWPAR models are its special cases (Remark 1). Let $\left(u_{ik},v_{ik},t_{k}\right)\left(i=1,2,\ldots ,n_{k},k=1,2,\ldots ,T\right)$ be any point in the studied spatiotemporal region. We fix a point $\left(u_{00},v_{00},t_{0}\right)$ and assume that $\beta_{jk}\left(u_{ik},v_{ik},t_{k}\right)\approx \beta_{j0}\left(u_{00},v_{00},t_{0}\right)\left(j=1,2,\ldots ,p\right)$. Then, the model (Equation 4) can be rewritten by
(5)$$\begin{array}{c} \eta_{ik}=g\left(\mu_{ik}\right)=\rho \sum_{m=1}^{T}\sum_{l=1}^{n_{k}}c_{lm}^{\left(ik\right)}\eta_{lm}+\sum_{j=1}^{p}\sum_{k=1}^{T}\beta_{j0}\left(u_{00},v_{00},t_{0}\right)X_{ijk}. \end{array}$$

Denote $\beta \left(u_{00},v_{00},t_{0}\right)={\left(\beta_{10},\ldots ,\beta_{p0}\right)}^{\prime }$, $X=diag(X_{i.})$ and $X_{i\cdot }=\left(X_{i1},\ldots ,X_{ip}\right)$. The corresponding matrix form can be represented as $\eta =\rho C\eta +\beta^{\prime }\left(u_{00},v_{00},t_{0}\right)X^{\prime }$.

### 3.1. Estimation of Parameter Vector β

For the fixed point $\left(u_{00},v_{00},t_{0}\right)$, we define a spatiotemporal distance $d_{ik}^{\left(0\right)}$ from this point to $\left(u_{ik},v_{ik},t_{k}\right)$ as $d_{ik}^{\left(0\right)}=\sqrt{\lambda \left[{\left(u_{00}-u_{ik}\right)}^{2}+{\left(v_{00}-v_{ik}\right)}^{2}\right]+\mu {\left(t_{0}-t_{k}\right)}^{2}}.$ The Gauss kernel function of these two points can be written by
$$\begin{array}{ccc} w_{ik}\left(u_{00},v_{00},t_{0}\right) & = & \frac{1}{\sqrt{2\pi }}\exp\left\{-\frac{1}{2}{\left(\frac{d_{ik}^{\left(0\right)}}{h_{ST}}\right)}^{2}\right\}\\ & = & \frac{1}{\sqrt{2\pi }}\exp\left\{-\frac{1}{2}\frac{\lambda \left[{\left(u_{00}-u_{ik}\right)}^{2}+{\left(v_{00}-v_{ik}\right)}^{2}\right]+\mu {\left(t_{0}-t_{k}\right)}^{2}}{h_{ST}^{2}}\right\}\\ & = & \frac{1}{\sqrt{2\pi }}\exp\left\{-\frac{1}{2}\left(\frac{{\left(u_{00}-u_{ik}\right)}^{2}+{\left(v_{00}-v_{ik}\right)}^{2}}{h_{S}^{2}}+\frac{{\left(t_{0}-t_{k}\right)}^{2}}{\tau h_{S}^{2}}\right)\right\}, \end{array}$$
where $h_{ST}$ and $h_{S}$ are the space-time bandwidth and space bandwidth, respectively. Meanwhile, we have $h_{ST}^{2}=\lambda h_{S}^{2}$, and τ=λ/μ is a spatiotemporal factor. Without loss of generality, let λ=1. Then, the weighted maximum likelihood of $Y_{ik}\left(i=1,2,\ldots ,n_{k},k=1,2,\ldots ,T\right)$ at the point $\left(u_{00},v_{00},t_{0}\right)$ is
$$\begin{array}{ccc} & & L\left(\beta_{10},\beta_{20},\cdots ,\beta_{p0}\right)=\prod_{k=1}^{T}\prod_{i=1}^{n_{k}}f\left(y_{ik};\theta_{ik},\phi_{ik}\right)w_{ik}\left(u_{00},v_{00},t_{0}\right), \end{array}$$
where $f(y_{ik};\theta_{ik},\phi_{ik})$ is the density function. The log-likelihood can be obtained as follows: $$\begin{array}{c} L_{1}\left(\beta \left(u_{00},v_{00},t_{0}\right)\right)=\sum_{k=1}^{T}\sum_{i=1}^{n_{k}}\left(\frac{y_{ik}\theta_{ik}-b\left(\theta_{ik}\right)}{a(\phi_{ik})}+c\left(y_{ik},\phi_{ik}\right)\right)w_{ik}\left(u_{00},v_{00},t_{0}\right). \end{array}$$

Note that $c\left(y_{ik},\phi_{ik}\right)=-\ln\left(y_{ik}!\right)$, $b\left(\theta_{ik}\right)=\mu_{ik}=\exp\left(\theta_{ik}\right)$, and $a\left(\phi_{ik}\right)=\phi_{ik}=1$. Thus, $E\left(Y_{ik}\right)=b^{\prime }\left(\theta_{ik}\right)=\exp\left(\theta_{ik}\right)=\mu_{ik},Var\left(Y_{ik}\right)=b^{\prime \prime }\left(\theta_{ik}\right)a\left(\phi_{ik}\right)=\exp\left(\theta_{ik}\right)=\mu_{ik}.$ Differentiating $L_{1}$ with respect to $\beta (u_{00},v_{00},t_{0})$ yields
(6)$$\begin{array}{c} \frac{\partial L_{1}}{\partial \beta_{r0}}=\sum_{k=1}^{T}\sum_{i=1}^{n_{k}}\left(\frac{y_{ik}-\mu_{ik}}{a_{ik}\phi }\frac{\partial \theta_{ik}}{\partial \beta_{r0}}\right)w_{ik}\left(u_{00},v_{00},t_{0}\right)=0, \end{array}$$
where $\beta_{r0}=\beta_{r}\left(u_{00},v_{00},t_{0}\right)\left(r=1,2,\cdots ,p\right)$, and
$$\begin{array}{c} \frac{\partial \theta_{ik}}{\partial \beta_{r0}}={\left(\frac{\partial \mu_{ik}}{\partial \theta_{ik}}\right)}^{-1}\frac{\partial \mu_{ik}}{\partial g(\mu_{ik})}\frac{\partial g(\mu_{ik})}{\partial \beta_{r0}}=\frac{1}{b^{\prime \prime }\left(\theta_{ik}\right)}\frac{1}{g^{\prime }\left(\mu_{ik}\right)}\frac{\partial \eta_{ik}}{\partial \beta_{r0}}. \end{array}$$

For convenience, let $N=\sum_{k=1}^{N}n_{k}$ and $W={\left(w_{ik}\left(u_{00},v_{00},t_{0}\right)\right)}_{N\times N}$. Denote $A={\left(I_{N}-\rho C\right)}^{-1}$, $Y={\left(Y_{11},\cdots ,Y_{n_{1}1},\cdots ,Y_{1T},\cdots ,Y_{n_{T}T}\right)}^{{}^{\prime }}$, $\mu ={\left(\mu_{11},\cdots ,\mu_{n_{1}1},\cdots ,\mu_{1T},\cdots ,\mu_{n_{T}T}\right)}^{{}^{\prime }}$, $\theta ={\left(\theta_{11},\cdots ,\theta_{n_{1}1},\cdots ,\theta_{1T},\cdots ,\theta_{n_{T}T}\right)}^{{}^{\prime }}$, $\phi ={\left(\phi_{11},\cdots ,\phi_{n_{1}1},\cdots ,\phi_{1T},\cdots ,\phi_{n_{T}T}\right)}^{{}^{\prime }}$. Suppose that $Y_{K}$, $\mu_{K}$, $\theta_{K}$ and $\phi_{K}$ are the *K*th elements of *Y*, μ, θ and ϕ, respectively. Then, we take the derivative of the model (Equation 5) with respect to $\beta_{r0}$, and obtain
$$\begin{array}{c} \frac{\partial \eta_{l}}{\partial \beta_{r0}}=\sum_{h=1}^{N}A_{lh}X_{hr}=A_{l\cdot }X_{\cdot r},\;l=1,2,\ldots ,N. \end{array}$$
The calculation process is given in Section A.2. Thus, the Equation (Equation 6) can be rewritten as
$$\begin{array}{c} \frac{\partial L_{1}}{\partial \beta_{r0}}=\frac{1}{\phi }\sum_{l=1}^{N}T_{l}A_{l\cdot }X_{\cdot r}\left(Y_{l}-\mu_{l}\right)g^{\prime }\left(\mu_{l}\right)W_{l}\left(u_{00},v_{00},t_{0}\right)=0. \end{array}$$

However, there is not a close-form solution for $\beta (u_{00},v_{00},t_{0})$. The Newton–Raphson procedure and Fisher scoring algorithm are used to get the estimation of β. The iterative formula is expressed as
(7)$$\begin{array}{ccc} {\hat{\beta }}^{\left(m+1\right)}\left(u_{00},v_{00},t_{0}\right) & = & {\hat{\beta }}^{\left(m\right)}\left(u_{00},v_{00},t_{0}\right)+I^{-1}\left({\hat{\beta }}^{\left(m\right)}\left(u_{00},v_{00},t_{0}\right)\right)S\left({\hat{\beta }}^{\left(m\right)}\left(u_{00},v_{00},t_{0}\right)\right)\\ & = & {\left({\left(A^{\left(m\right)}X\right)}^{\prime }T^{\left(m\right)}W\left(u_{00},v_{00},t_{0}\right)\left(A^{\left(m\right)}X\right)\right)}^{-1}\\ & & \times {\left(A^{\left(m\right)}X\right)}^{\prime }T^{\left(m\right)}W\left(u_{00},v_{00},t_{0}\right)Z^{\left(m\right)}, \end{array}$$
where the Fisher information matrix I(β)=E(I(β)), and
$$S\left({\hat{\beta }}^{\left(m\right)}\left(u_{00},v_{00},t_{0}\right)\right)={\left(\frac{\partial L_{1}}{\partial \beta_{10}},\frac{\partial L_{1}}{\partial \beta_{20}},\cdots ,\frac{\partial L_{1}}{\partial \beta_{p0}}\right)}^{\prime }$$
is the scalar vector. The detail process is provided in Section A.2. For the fixed point $\left(u_{ik},v_{ik},t_{k}\right)\left(i=1,2,\ldots ,n_{k};k=1,2,\ldots ,T\right)$, ${\hat{\beta }}_{jk}\left(u_{ik},v_{ik},t_{k}\right)$ can be obtained by (Equation 7).

**Remark** **2.**
*The estimations $\hat{\beta }\left(u_{ik},v_{ik},t_{k}\right)\left(i,l=1,2,\ldots ,n_{k},k,m=1,2,\ldots ,T\right)$ are related to the temporal and spatial effects in the GTWPAR model. If m≠k, $w_{ik}\left(u_{lm},v_{lm},t_{m}\right)=0$ and $c_{lm}^{\left(ik\right)}=0$, then $\hat{\beta }\left(u_{ik},v_{ik},t_{k}\right)=\hat{\beta }\left(u_{i},v_{i}\right)$ correspond to the parameter estimations of the GWPAR model. If $w_{ik}\left(u_{lm},v_{lm},t_{m}\right)=0\left(m\neq k\right)$ and C=0, they are the estimations of the GWPR model. If W=0 and C=0, then $\hat{\beta }\left(u_{ik},v_{ik},t_{k}\right)=\hat{\beta }$ are the global estimation values of the PR model.*


### 3.2. Estimation of Parameter ρ

Based on the density function, the log-likelihood function of ρ is
$$\begin{array}{c} L_{2}\left(\rho \right)=\sum_{k=1}^{T}\sum_{i=1}^{n_{k}}\left(\frac{y_{ik}\theta_{ik}-b\left(\theta_{ik}\right)}{a(\phi_{ik})}+c\left(y_{ik},\phi_{ik}\right)\right). \end{array}$$
Differentiating $L_{2}\left(\rho \right)$ with respect to ρ, we have
(8)$$\begin{array}{c} \frac{\partial L_{2}}{\partial \rho }=\sum_{k=1}^{T}\sum_{i=1}^{n_{k}}\left(\frac{y_{ik}-\mu_{ik}}{a_{ik}\phi }\frac{\partial \theta_{ik}}{\partial \rho }\right)=0, \end{array}$$
where $\frac{d\theta_{ik}}{d\rho }=\frac{1}{b^{\prime \prime }\left(\theta_{ik}\right)g^{\prime }\left(\mu_{ik}\right)}\frac{d\eta_{ik}}{d\rho }.$ Then, we take the derivative of the model (Equation 5) with respect to ρ as follows:$$\begin{array}{c} \frac{d\eta_{l}}{d\rho }=\frac{dg(\mu_{l})}{d\rho }=\sum_{h=1}^{N}A_{l\cdot }C_{\cdot h}\eta_{h}. \end{array}$$
The detail calculation is given in Section A.3. Then, the Equation (Equation 8) can be rewritten in the following nonlinear form
$$\frac{dL_{2}}{d\rho }=\sum_{l=1}^{N}\frac{\left(Y_{l}-\mu_{l}\right)\sum_{h=1}^{N}A_{l\cdot }C_{\cdot h}\eta_{h}}{a_{l}\phi V\left(\mu_{l}\right)g^{\prime }\left(\mu_{l}\right)}=0.$$

According to the Newton–Raphson procedure and Fisher scoring algorithm, the iterative formula of ${\hat{\rho }}^{\left(m+1\right)}$ is
(9)$$\begin{array}{ccc} {\hat{\rho }}^{\left(m+1\right)} & = & {\hat{\rho }}^{\left(m\right)}+I^{-1}\left({\hat{\rho }}^{\left(m\right)}\right)S\left({\hat{\rho }}^{\left(m\right)}\right)\\ & = & {\hat{\rho }}^{\left(m\right)}+{\left({\left(A^{\left(m\right)}C\eta^{\left(m\right)}\right)}^{\prime }T^{\left(m\right)}\left(A^{\left(m\right)}C\eta^{\left(m\right)}\right)\right)}^{-1}\\ & & \times {\left(A^{\left(m\right)}C\eta^{\left(m\right)}\right)}^{\prime }T^{\left(m\right)}\left(Z^{\left(m\right)}-\eta^{\left(m\right)}\right), \end{array}$$
where the scalar vector $S\left({\hat{\rho }}^{\left(m\right)}\right)=\frac{1}{\phi }{\left(AC\eta \right)}^{\prime }T\left(Z-\eta \right)$ and the Fisher information matrix $I\left(\rho \right)=\frac{1}{\phi }{\left(AC\eta \right)}^{\prime }T\left(AC\eta \right)$. The calculation process of the scalar vector $S({\hat{\rho }}^{\left(m\right)})$ and the information matrix I is given in Section A.3.

## 4. Main Results

In this section, we apply the PR, GWPR, GWPAR and GTWPAR models to analyze the relationships between the HIV/AIDS epidemic, the GDP per capita and population density in ten countries of Sub-Saharan Africa from 2011 to 2016. The ten countries are Angola, Botswana, Lesotho, Malawi, Mozambique, Namibia, South Africa, Swaziland, Zimbabwe and Zambia. The parameters of these four models are estimated by the Newton–Raphson procedure and Fisher scoring algorithm. The coefficient of determination R${}^{2}$, the corrected Akaike information criterion (AICc), the deviation (D) and mean-square error (MSE) are used to compare the performances of the four models [18].

### 4.1. The HIV/AIDS Epidemic Models

The data of HIV/AIDS incidence, GDP per capita and population density were derived from http://data.cnki.net/InternationalData/Report. Readers should note that authorization is required to access the database on this website. Figure 1 describes the HIV/AIDS incidence in ten countries from 2011 to 2016. It shows that the incidence varies significantly in different regions. Angola has a minimum incidence of less than 5%, while Botswana and Swaziland have higher incidences of more than 20% every years. Therefore, it may be necessary to consider the temporal and spatial factors in analyzing the HIV/AIDS epidemic.

The distributions of HIV/AIDS cases, GDP per capita and population density are displayed in Figure 2. The Pearson correlation coefficients between these cases and GDP per capita and population density are 0.2739 and −0.1179, respectively. Meanwhile, the two socioeconomic factors have different effects on the HIV/AIDS cases at the spatiotemporal locations. These reflect a spatiotemporal non-stationarity between the cases and two factors in ten countries from 2011 to 2016. Table 1 lists the *p*-values of the first-order autocorrelation of HIV/AIDS cases in the different years of the same region or the different regions of the same year. Each region has a significant spatial autocorrelation (*p*-value < 0.01) each year. Lesotho and South Africa had temporal autocorrelation during 2011 to 2016. Thus, the spatial and temporal autocorrelation should not be ignored.

Next, we standardized the two socioeconomic factors. The multiplex collinear test [32] was performed by the condition number $k=\sqrt{\lambda_{\max}/\lambda_{\min}}=1.804\left(\leq 15\right)$ (λ is the eigenvalue of explanatory variable matrix). If k>15, then the data have collinearity. Otherwise, there is no collinearity. Thus, there is no collinearity between the two factors. Let $\mu_{ik}$, $r_{ik}$ and $P_{ik}$ be the annual HIV/AIDS cases (Unit: 1/1000 people), incidence (Unit: 1/100) and total population (unit: 100,000 people) in the *k*th year of the *i*th region, respectively. Denote $g\left(\mu_{ik}\right)=\eta_{ik}=\ln\mu_{ik}=\ln r_{ik}+\ln P_{ik}(\mu_{ik}=r_{ik}P_{ik},i=1,2,\ldots ,10$ and k=1,2,…,6). Let $X_{i1k}$ and $X_{i2k}$ be the GDP per capita and population density in the *i*th region at the *k*th year, respectively. The PR model is written by
(10)$$\begin{array}{c} g\left(\mu_{ik}\right)=\beta_{0}+\beta_{1}X_{i1k}+\beta_{2}X_{i2k},\;i=1,2,\ldots ,10,k=1,2,\ldots ,6, \end{array}$$
where $\beta_{j}\left(j=0,1,2\right)$ are unknown constants. The GWPR model is introduced as
(11)$$\begin{array}{c} g\left(\mu_{ik}\right)=\beta_{0}\left(u_{ik},v_{ik}\right)+\beta_{1}\left(u_{ik},v_{ik}\right)X_{i1k}+\beta_{2}\left(u_{ik},v_{ik}\right)X_{i2k},\;i=1,2,\ldots ,10, \end{array}$$
where *k* is a fixed constant taken from {1,2,…,6}, and $\beta_{j}\left(u_{ik},v_{ik}\right)$ are unknown spatial parameters for the *i*th country $\left(u_{ik},v_{ik}\right)$ in the *k*th year. Let ρ be a scalar autoregressive parameter, and $c_{il}$ be a constant that represents an adjacency relation. The GWPAR model is
(12)$$\begin{array}{c} g\left(\mu_{ik}\right)=\rho \sum_{l=1}^{n}c_{il}\eta_{ik}+\beta_{0}\left(u_{ik},v_{ik}\right)+\beta_{1}\left(u_{ik},v_{ik}\right)X_{i1k}+\beta_{2}\left(u_{ik},v_{ik}\right)X_{i2k}, \end{array}$$
where n=10, *k* is a fixed constant, and $\beta_{j}\left(u_{ik},v_{ik}\right)$ are defined as above. Let $c_{lm}^{\left(ik\right)}$ be a spatiotemporal adjacency relation, and $\beta_{jk}\left(u_{ik},v_{ik},t_{k}\right)\left(k=1,2,\ldots ,6\right)$ be unknown spatiotemporal parameters in the *i*th country $\left(u_{ik},v_{ik}\right)$ in the *k*th year. The GTWPAR model is established as follows:(13)$$\begin{array}{ccc} g(\mu_{ik}) & = & \rho \sum_{m=1}^{T}\sum_{l=1}^{n_{k}}c_{lm}^{\left(ik\right)}\eta_{lm}+\beta_{0k}\left(u_{ik},v_{ik},t_{k}\right)\\ & & +\beta_{1k}\left(u_{ik},v_{ik},t_{k}\right)X_{i1k}+\beta_{2k}\left(u_{ik},v_{ik},t_{k}\right)X_{i2k}, \end{array}$$
where T=6; $n_{k}=10$ for every *k* years; and ρ is defined as above.

Algorithms I, II, III and IV of PR, GWPR, GWPAR and GTWPAR models are provided in Section A.4, respectively.

### 4.2. Statistical Analysis

For the PR model, we get the estimated values of unknown parameters by Algorithm I. Then, the best space bandwidth is chosen by the cross-validation method. Following Huang et al. [28], the range [0.09, 2.49] of the space bandwidth is selected according to the minimum and maximum distance of the geographical positions. In the GWPR model, the best space bandwidth is h=0.62,0.59,0.62,0.61,0.60,0.60, and the estimations of coefficient functions are given by Algorithm II. The optimal space bandwidth of the GWPAR model is selected as h=1.2895,1.1316,1.1316,1.0526,1.0526,1.0526. Based on Algorithm III, we can get the estimations of coefficient functions and the scalar autoregressive parameter $\hat{\rho }=0.267,0.269,0.263,0.264,0.264,0.264.$ For the GTWPAR model, we chose $h_{s}=1.1316,0.9737,0.8947,0.9211,0.7842,0.8789$ and τ=0.1, where τ(>0) is a balanced parameter. The coefficient estimations and scalar autoregressive parameter $\hat{\rho }=0.126$ can be obtained by Algorithm IV. The quantile and mean values of coefficient estimations and response variables are shown in Table 2. We note that the GWPR, GWPAR and GTWPAR models can reflect the non-stationarity property of the influencing factors; the PR model cannot. Moreover, the GTWPAR model has a better performance than other models by comparing the true and fitted values.

The average estimated coefficients are visualized in Figure 3. For the PR model, the GDP per capita and population density had the same effect on the HIV/AIDS epidemic for ten countries in six years. However, there exist significant spatial non-stationarity and auto-correlation for different countries under the GWPR, GWPAR and GTWPAR models. Figure 4 shows the spatial distribution of the average MSE of their response variables. The lighter the color, the smaller the average error is. Thus, the GWPAR and GTWPAR models have the better fitting results.

These four indicators can effectively compare the performances of the proposed models (Table 3). The calculation formulas of R${}^{2}$, AICc, D and MSE are given in Section A.5. The coefficient of determination R${}^{2}$ gradually increases from 12.91% of the PR model to 99.57% of the GTWPAR model. The MSE, AICc and D values of the GTWPAR model are smaller than those of other models. Therefore, the GTWPAR model is more suitable to investigate the spatiotemporal HIV/AIDS epidemic.

Based on the GTWPAR model, the mean values and 95% confidence intervals of the coefficient estimations are shown in Figure 5. The mean estimations are represented by the dot, and the 95% confidence intervals are given by the upper and lower lines. Note that the GDP per capita in Botswana, Namibia and South Africa has a positive effect on the HIV/AIDS cases. Six other countries (except Lesotho) had the opposite results. The population density for five countries had a positive effect on the HIV/AIDS cases—Angola, Botswana, Namibia, South Africa and Zambia. The population density of other five countries had the negative effect. Moreover, the impact of the GDP per capita on HIV/AIDS epidemic had a strong spatiotemporal non-stationarity in Lesotho, Malawi and Zimbabwe, while the population density had a strong spatiotemporal non-stationarity in Angola.

## 5. Conclusions

In this paper, we propose four regression models, including the PR, GWPR, GWPAR and GTWPAR, to investigate the non-stationary and auto-correlation properties. The relationships between the HIV/AIDS epidemic, GDP per capita and population density were analyzed in ten countries of Sub-Saharan Africa from 2011 to 2016. The unknown parameters of these models can be estimated by the Newton–Raphson procedure and Fisher scoring algorithm.

The PR model is a classical generalized model, which considers the global relationships between the response and explanatory variables. The GWPR and GWPAR models have been introduced to determine the spatial non-stationarity or auto-correlation. The GTWPAR model proposed by this article can be used to investigate not only spatiotemporal non-stationary but also auto-correlation. Thus, the PR, GWPR and GWPAR models are several special cases of the GTWPAR model (see Remark 1 and Remark 2). The performances of these models were evaluated by analyzing the correlations between the HIV/AIDS epidemic and two socioeconomic factors. The parameter estimations of the models can be obtained by Algorithms I, II, III and IV in Section A.4.

The results show that the impacts of GDP per capita and population density on HIV /AIDS cases had significant spatiotemporal non-stationarity and auto-correlation. The GWPR, GWPAR and GTWPAR models can reflect the strong spatial or spatiotemporal non-stationarity. The auto-correlation can be reflected in the GWPAR and GTWPAR models. Compared with other models, the GTWPAR model is more effective in terms of four comparison indicators. Thus, we suggest that the GTWPAR model can be used to analyze the spatiotemporal characteristics of the HIV/AIDS epidemic and the influences of the GDP per capita and population density.

Further work also exists in our study. For example, we observed that the effects of the GDP per capita for Lesotho, Malawi and Zimbabwe and the population density for Angola on HIV/AIDS had strong spatiotemporal non-stationarity. These may be the result of local environmental or political factors. Whether the fitting results of these regions will perform better if explanatory variables such as local unique environmental or political factors are added needs to be further investigated.

## Figures and Tables

**Figure 1 entropy-22-01230-f001:**
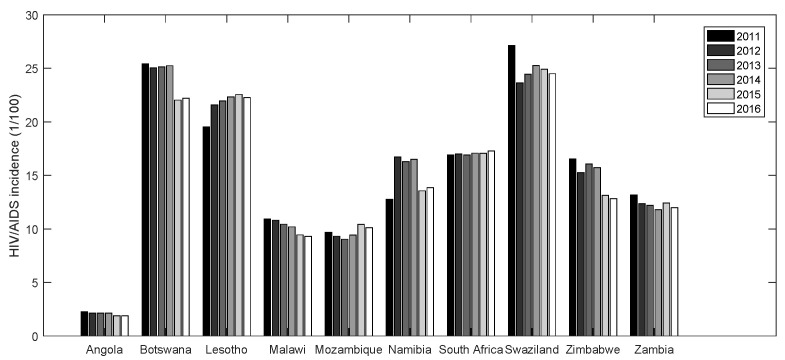
Spatiotemporal HIV/AIDS incidence of ten countries, 2011–2016.

**Figure 2 entropy-22-01230-f002:**
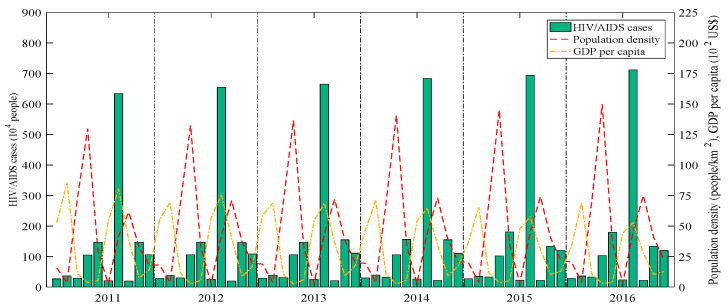
Distributions of HIV/AIDS cases, GDP per capita and population density of ten countries, 2011–2016.

**Figure 3 entropy-22-01230-f003:**
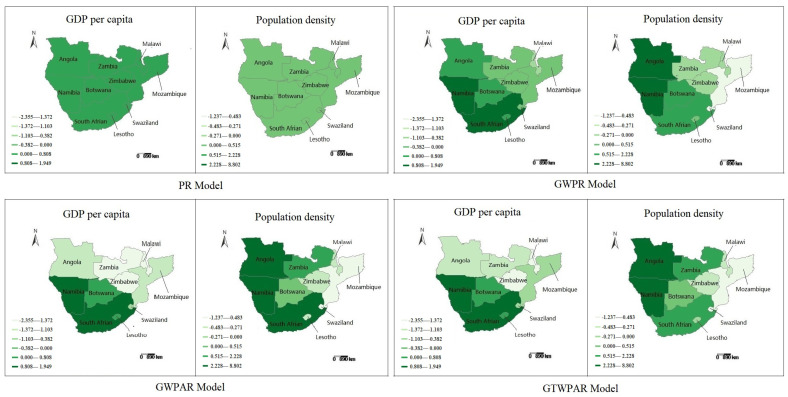
The spatial distribution of the average coefficient estimations in four models.

**Figure 4 entropy-22-01230-f004:**
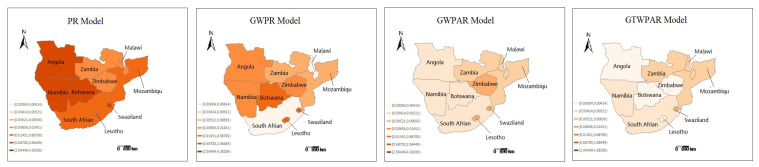
The average MSE of response variables.

**Figure 5 entropy-22-01230-f005:**
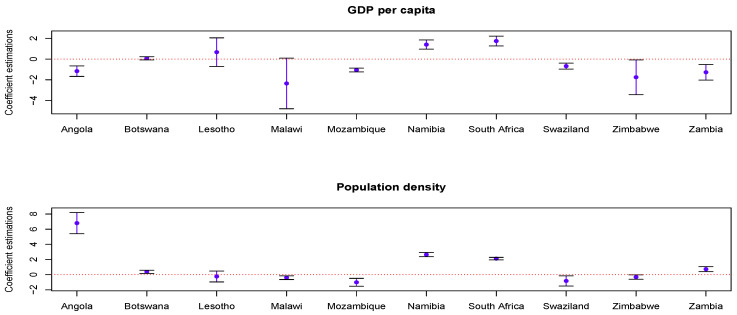
The mean values and 95% confidence intervals of coefficient estimations.

**Table 1 entropy-22-01230-t001:** *p*-values of the spatial and temporal autocorrelation analysis.

Time	2011	2012	2013	2014	2015	2016
*p*-value	0.0024	0.0028	0.0021	0.0019	0.0017	0.0022
**Regions**	**Angola**	**Botswana**	**Lesotho**	**Malawi**	**Mozambique**	**Namibia**
*p*-value	0.9094	0.8807	0.0092	0.5300	0.1289	0.8267
	**South Africa**	**Swaziland**	**Zimbabwe**	**Zambia**		
	0.0045	0.1491	0.5688	0.2231		

**Table 2 entropy-22-01230-t002:** The quantile and mean values of coefficient estimations and response variables.

Model	Coefficient	Min	1st Qu	Median	3rd Qu	Max	Mean
True	η	5.293	5.644	6.455	7.288	8.871	6.559
PR	${\hat{\beta }}_{1}$	0.581	0.581	0.581	0.581	0.581	0.581
	${\hat{\beta }}_{2}$	0.385	0.385	0.385	0.385	0.385	0.385
	$\hat{\eta }$	6.344	6.527	7.185	7.447	8.162	7.078
GWPR	${\hat{\beta }}_{1}$	−0.519	−0.249	0.009	0.641	1.774	0.288
	${\hat{\beta }}_{2}$	−0.672	−0.139	0.068	2.044	3.803	0.747
	$\hat{\eta }$	5.618	6.422	7.022	7.202	8.813	6.932
GWPAR	${\hat{\beta }}_{1}$	−6.286	−1.258	−0.910	0.240	3.036	−0.556
	${\hat{\beta }}_{2}$	−2.089	−0.669	0.019	2.281	9.152	0.912
	$\hat{\eta }$	5.176	5.625	6.435	7.219	8.940	6.539
GTWPAR	${\hat{\beta }}_{1}$	−5.642	−0.956	−0.677	0.186	2.992	−0.443
	${\hat{\beta }}_{2}$	−1.865	−0.464	0.215	2.134	9.104	0.988
	$\hat{\eta }$	5.200	5.589	6.387	7.215	8.803	6.489

**Table 3 entropy-22-01230-t003:** The comparison of the four models.

Model	R${}^{2}$	AICc	D	MSE
PR	0.1291	42,504.40	42,624.07	1.6488
GWPR	0.6139	6495.12	6613.02	0.4326
GWPAR	0.9940	155.46	236.02	0.0067
GTWPAR	0.9957	115.25	190.05	0.0048

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
