# Peer review of "Analysis of HIV/AIDS Epidemic and Socioeconomic Factors in Sub-Saharan Africa"

_entropy, 2020, doi:10.3390/e22111230_

Round 1

Reviewer 1 Report

Dear authors,

- More revision of references is recomended.

- It is necessary a deeper contextualization around the time series components that should characterize the data, describing and resalting the applicability of the proposed models.

- Time series Analysis often starts with a deep statistical description, that is a lack in the document.

Reviewer 2 Report

My report in the attached file.

Reviewer 3 Report

See my report...

Round 2

Reviewer 3 Report

I believe my concerns were addressed already  and  the present form of the paper is looking good to be published in entropy.